# Amino Acid-Metabolizing Enzymes in Advanced High-Grade Serous Ovarian Cancer Patients: Value of Ascites as Biomarker Source and Role for IL4I1 and IDO1

**DOI:** 10.3390/cancers15030893

**Published:** 2023-01-31

**Authors:** Yvonne Grobben, Judith E. den Ouden, Cristina Aguado, Anne M. van Altena, Aletta D. Kraneveld, Guido J. R. Zaman

**Affiliations:** 1Oncolines B.V., 5349 AB Oss, The Netherlands; 2Radboud Institute for Health Sciences, Radboud University Medical Center, Obstetrics and Gynecology, 6525 GA Nijmegen, The Netherlands; 3Laboratory of Oncology, Pangaea Oncology, Dexeus University Hospital, 08028 Barcelona, Spain; 4Division of Pharmacology, Faculty of Science, Utrecht Institute for Pharmaceutical Sciences, Utrecht University, 3584 CG Utrecht, The Netherlands

**Keywords:** ovarian cancer, immunotherapy, immunosuppression, liquid biopsy, ascites, metabolomics, amino acid metabolism, tryptophan, IL4I1, IDO1

## Abstract

**Simple Summary:**

Ovarian cancer is the most lethal gynecological malignancy in the United States. Despite the success of immunotherapy for treatment of various cancer types, its impact on ovarian cancer is restrained by a highly immunosuppressive tumor microenvironment. We aimed to evaluate the contribution of several amino acid-metabolizing enzymes to this environment by measuring the levels of amino acids and corresponding metabolites in liquid biopsies of high-grade serous ovarian cancer patients. The levels of different amino acid-derived metabolites were higher in ascites compared to plasma samples, demonstrating the value of utilizing ascites for biomarker identification. Moreover, the enzymes IDO1 and IL4I1 were identified as active players in high-grade serous ovarian cancer, and a correlation between IL4I1 metabolite levels and disease stage was revealed. Further exploration of the implications of enhanced IL4I1 activity in ovarian cancer is warranted to pave the way for new immunotherapeutic strategies in the treatment of this disease.

**Abstract:**

The molecular mechanisms contributing to immune suppression in ovarian cancer are not well understood, hampering the successful application of immunotherapy. Amino acid-metabolizing enzymes are known to contribute to the immune-hostile environment of various tumors through depletion of amino acids and production of immunosuppressive metabolites. We aimed to collectively evaluate the activity of these enzymes in high-grade serous ovarian cancer patients by performing targeted metabolomics on plasma and ascites samples. Whereas no indication was found for enhanced l-arginine or l-glutamine metabolism by immunosuppressive enzymes in ovarian cancer patients, metabolism of l-tryptophan by indoleamine 2,3-dioxygenase 1 (IDO1) was significantly elevated compared to healthy controls. Moreover, high levels of l-phenylalanine- and l-tyrosine-derived metabolites associated with interleukin 4 induced 1 (IL4I1) activity were found in ovarian cancer ascites samples. While l-tryptophan is a major substrate of both IDO1 and IL4I1, only its enhanced conversion into l-kynurenine by IDO1 could be detected, despite the observed activity of IL4I1 on its other substrates. In ascites of ovarian cancer patients, metabolite levels were higher compared to those in plasma, demonstrating the value of utilizing this fluid for biomarker identification. Finally, elevated metabolism of l-phenylalanine and l-tyrosine by IL4I1 correlated with disease stage, pointing towards a potential role for IL4I1 in ovarian cancer progression.

## 1. Introduction

Ovarian cancer is the fifth most lethal malignancy in women in the United States, and is the deadliest among cancers of the female reproductive system [1]. Patients with epithelial ovarian cancer, accounting for about 90% of all cases, are often diagnosed at advanced-stage disease due to the presentation of merely vague and nonspecific symptoms [2]. At these stages, the prognosis for patients is poor, culminating in a five-year survival rate below 50% for all stages combined [2,3].

For over the last two decades, the standard first-line treatment for advanced-stage epithelial ovarian cancer patients has been debulking surgery combined with platinum- and taxane-based combination chemotherapy. In more recent years, clinical trials focused on optimization of surgical and chemotherapy regimens, and Food and Drug Administration (FDA) approval of poly(ADP-ribose) polymerase (PARP) inhibitors and the antiangiogenic drug bevacizumab as frontline maintenance therapies have advanced patient treatment [4]. Nonetheless, the prognosis for ovarian cancer patients remains grim, particularly due to the high rate of disease recurrence as a result of drug resistance [5].

For several malignancies, the emergence of immune checkpoint inhibitors aimed at reactivating the anticancer immune response has prompted reshaping of treatment strategies [6,7]. Unlike classic cytotoxic chemotherapies, which act directly on tumor cells by inducing cell killing, immune checkpoint inhibitors disrupt inhibitory signaling between tumor and immune cells. However, despite their prominent success in different cancer types, clinical trials evaluating immune checkpoint inhibitors have demonstrated limited efficacy in ovarian cancer patients [8]. A low tumor mutational burden, associated with the production of fewer immunogenic neoantigens, and a highly immunosuppressive tumor microenvironment (TME) may underly these clinical observations [9].

Several amino acid-metabolizing enzymes have been implicated in the attenuation of antitumor immune responses (Figure 1), either through depletion of amino acids from the TME or the production of immunosuppressive metabolites. Expression of glutaminase 1 (GLS1) is frequently elevated in malignant cells as a result of their metabolic reprogramming [10], while arginase 1 (ARG1) is secreted from myeloid cells in patients with various cancer types [11]. GLS1 and ARG1 can deprive the TME of l-glutamine (Gln) and l-arginine (Arg), respectively, thereby restraining effector T cell proliferation and functionality [10,11]. Alternatively, aberrant metabolism of Arg by tumor or myeloid cell-expressed inducible nitric oxide synthase (iNOS) can yield high concentrations of the small molecule nitric oxide (NO), which has various immunosuppressive properties [12].

Indoleamine 2,3-dioxygenase 1 (IDO1), expressed by both tumor and immune cells, is the most extensively studied amino acid-metabolizing enzyme relevant to cancer immunology. IDO1 exerts its immunosuppressive effects through local depletion of l-tryptophan (Trp) as well as generation of Trp metabolites that act as agonists of the aryl hydrocarbon receptor (AhR) [13]. The AhR plays a central role in inducing tolerogenic immune responses [14]. Moreover, although considerably less studied, the frequently tumor or stromal cell-overexpressed tryptophan 2,3-dioxygenase (TDO) appears to act through similar mechanisms [15,16]. Most recently, interleukin 4 induced 1 (IL4I1), secreted by professional antigen-presenting cells in various cancer types [17], was indicated as yet another enzyme capable of producing AhR agonists through metabolism of Trp [18]. However, its immunosuppressive effects may also arise from the generation of hydrogen peroxide (H_2_O_2_) through metabolism of its other major substrates, l-phenylalanine (Phe) and l-tyrosine (Tyr) [19].

The enzymes described above have each been investigated as a potential drug target for cancer immunotherapy, mostly in combination with immune checkpoint blockade, but can also represent potential biomarkers for disease prognosis [10,11,12,13,18]. Through use of immunohistochemical methods, enhanced expression of amino acid-metabolizing enzymes has been demonstrated in tumor tissue biopsies of various cancer types [16,17,20,21]. In addition, enzymatic activity has been evaluated in plasma or serum specimens by direct measurement of secreted enzyme activity (in case of ARG1) or indirectly through analysis of amino acid and metabolite abundancy [21,22,23]. For ovarian cancer patients, the frequent presence of ascites (i.e., fluid build-up in the peritoneal cavity) presents another source of potential biomarkers for diagnostic, prognostic or predictive purposes. Ascites can be obtained through a substantially less invasive procedure compared to tumor biopsy and may provide a stronger indication of tumor-related enzyme activity compared to plasma or serum due to its proximity to the tumor site. A similar opportunity is presented in other types of cancer by the presence of pleural effusion, cerebrospinal fluid or urine as alternative liquid biopsy sources.

In the current study, we aimed to collectively evaluate the presence and role of the different immunosuppressive amino acid-metabolizing enzymes in high-grade serous ovarian cancer, the most common epithelial ovarian cancer subtype [2]. To this end, the abundance of relevant amino acids and metabolites as indirect markers for enzymatic activity were determined in liquid biopsies by targeted metabolomics using liquid chromatography–tandem mass spectrometry (LC-MS/MS). Moreover, by comparison of plasma and ascites specimens of ovarian cancer patients, we sought to determine the potential benefit of using ascites as a source of biomarkers related to these enzymes. Finally, we aimed to determine whether our findings could be extended beyond high-grade serous ovarian cancer by analysis of pleural effusion samples from non-small cell lung cancer patients.

## 2. Materials and Methods

### 2.1. Chemicals and Reagents

All analytes and isotope-labelled internal standards for LC-MS/MS analysis were obtained from commercial vendors as listed in Appendix A. UPLC-grade acetonitrile and methanol were purchased from VWR.

### 2.2. Patient and Healthy Control Samples

Blood and ascites were collected from patients with a primary diagnosis of advanced-stage (i.e., International Federation of Gynecology and Obstetrics (FIGO) stage IIb–IV) high-grade serous ovarian cancer at Radboud university medical center in Nijmegen, Canisius Wilhelmina Hospital in Nijmegen, Catharina Hospital in Eindhoven and Rijnstate Hospital in Arnhem (The Netherlands). Eligible patients were above eighteen years of age, presented with sufficient ascites for collection, had no previous or concurrent malignant disease, and were chemotherapy naïve. Patients who did not complete primary treatment were excluded from survival analysis. Pleural effusions were collected from patients with a primary diagnosis of advanced-stage (i.e., American Joint Committee on Cancer (AJCC) stage IV) non-small cell lung cancer at Dexeus University Hospital and Teknon Medical Center in Barcelona (Spain). Collection of the samples and the research described was conducted with approval of the medical ethical committees of the corresponding hospitals and informed written consent from each subject.

Blood from patients with high-grade serous ovarian cancer was collected in BD Vacutainer Lithium Heparin tubes (#367526; BD, Franklin Lakes, NJ, USA) and was stored at room temperature (RT) prior to processing. The day after collection, the blood was centrifuged for 10 min at 1500 rpm and the resultant plasma was stored at −80 °C or below. Ascites collected from the patients was stored at 2–8 °C until processing. The day after collection, the ascites was filtered through a 100 µm cell strainer and centrifuged for 10–15 min at 1500 rpm. The supernatant was stored at −80 °C. Pleural effusion from patients with non-small cell lung cancer was centrifuged for 10 min at 2300 rpm and the supernatant was stored at −20 °C.

Lithium heparin plasma samples from healthy donors were obtained from TCS Biosciences (Buckingham, UK). Blood from these donors was stored at 2–8 °C prior to processing within 24 h of collection. The age distribution of the healthy donors was matched as closely as possible to those of the ovarian cancer patients, but no exact match could be obtained due to the limited availability of donors with ages above 60 years.

Healthy donor blood samples for evaluation of the stability of amino acid and metabolite levels in whole blood were obtained from Sanquin (Nijmegen, The Netherlands) in BD Vacutainer Lithium Heparin tubes. Blood was aliquoted upon arrival from the provider. At different time points, plasma was separated and stored as described above.

### 2.3. Standard and Sample Preparation for LC-MS/MS Analysis

All analytes and internal standards were dissolved at 10 or 50 mM concentration in MilliQ water (MQ), with one or two equivalents of HCl or NaOH if required for solubility, or 100% DMSO and stored at −80 °C. For each LC-MS/MS experiment, eight calibration standards, four quality control samples and an internal standard mixture were prepared by dilution in MQ.

For hydrophilic interaction liquid chromatography (HILIC)-MS/MS analysis, 5 µL plasma, ascites, pleural effusion or 5% BSA in PBS was diluted with 50 µL standard solution, quality control solution or MQ in a 2 mL 96-well Masterblock^®^ plate (#780270; Greiner Bio-One, Kremsmünster, Austria). All samples were spiked with 45 µL internal standard mixture and agitated at 1500 rpm for 1 min. Extraction was performed with 400 µL acetonitrile and agitation at 1400 rpm for 1 min. The samples were centrifuged for 30 min at 4000 rpm at 4 °C and 300 µL supernatant was transferred to a 2 mL TrueTaper^®^ 96-well collection plate (#968820; Screening Devices, Amersfoort, The Netherlands) using a Hamilton Microlab STARlet liquid handler (Hamilton, Reno, NV, USA).

For reversed-phase liquid chromatography (RPLC)-MS/MS analyses, a surrogate matrix was prepared by incubating pooled healthy donor plasma with 60 mg activated charcoal per mL plasma for two hours, followed by centrifugation at 14,000 rpm for 10 min, according to the previous literature [24]. In a 2 mL 96-well Masterblock^®^ plate, 30 µL plasma, ascites, pleural effusion or surrogate matrix was diluted with 10 µL standard solution, quality control solution or MQ. All samples were spiked with 10 µL internal standard mixture and agitated at 1650 rpm for 1 min. Extraction was performed with 325 µL acetonitrile and agitation at 1200 rpm for 1 min. The samples were centrifuged for 60 min at 4000 rpm at 4 °C and 300 µL supernatant was transferred to a 1 mL TrueTaper^®^ 96-well plate (#968810; Screening Devices). The samples were evaporated for 42 min under a 50 °C nitrogen stream at 7 psi pressure using an Ultravap (Porvair Sciences, Norfolk, UK) with straight needle head and were subsequently dissolved in 60 µL 0.1% acetic acid in MQ. The samples were centrifuged for 60 min at 4000 rpm at 4 °C to precipitate undissolved components and 50 µL supernatant was transferred to a 2 mL TrueTaper^®^ 96-well collection plate. Plates were covered with a pre-slit silicone mat (#964085; Screening Devices) and kept in the autosampler at 10 °C until analysis.

Validation of the surrogate matrices (i.e., 5% BSA in PBS for HILIC-MS/MS and charcoal-stripped plasma for RPLC-MS/MS) was performed by evaluation of the absence of endogenous metabolite levels in these matrices and the standard curve parallelism between the surrogate matrices and all sample matrices (i.e., plasma, ascites and pleural effusion) using the method of standard addition [25].

### 2.4. Liquid Chromatography

Analyte separation was performed on an ACQUITY UPLC System (Waters Corporation, Milford, MA, USA). Arginine, ornithine, citrulline, glutamine, glutamic acid, phenylalanine, tyrosine, tryptophan and their respective internal standards were separated by HILIC on an ACQUITY UPLC BEH amide column (130 Å; 1.7 µM; 2.1 mm × 100 mm; #186004801; Waters) with 10 mM NH_4_HCO_2_ in MQ at pH 3.0 as mobile phase A and 0.1% formic acid in acetonitrile as mobile phase B. Elution was performed after injection of 2 µL sample at a flow rate of 0.4 mL/min and a column temperature of 35 °C using the following gradient: 0–0.5 min, 80% B; 0.5–5 min, 80–79% B; 5–7 min, 79–60% B; 7–8 min, 60–50% B; 8–9 min, 50% B; 9–10 min, 50–80% B; followed by 10 min at 80% B for column re-equilibration.

The remaining analytes and their internal standards were separated by RPLC on an ACQUITY UPLC HSS T3 column (100 Å; 1.8 µM; 2.1 mm × 50 mm; #186003538; Waters) at a flow rate of 0.6 mL/min and a column temperature of 25 °C with three different gradients. Kynurenine, kynurenic acid, indole-3-lactic acid, indole-3-acetic acid and their respective internal standards were separated after 5 µL injection using 0.1% formic acid in MQ as mobile phase A and 0.1% formic acid in acetonitrile as mobile phase B with the following gradient: 0–0.5 min, 1–18% B; 0.5–2.5 min, 18–31% B; 2.5–2.6 min, 31–100% B; 2.6–3.5 min, 100% B; 3.5–3.6 min, 100–1% B; 3.6–4 min, 1% B (method RPLC-1). Indole-3-aldehyde and its internal standard were separated after 5 µL injection using the same mobile phases with the following gradient: 0–0.25 min, 1–20% B; 0.25–2 min, 20–32% B; 2–2.1 min, 32–100% B; 2.1–3 min, 100% B; 3–3.1 min, 100–1% B; 3.1–3.5 min, 1% B (method RPLC-2). Phenylpyruvic acid, 4-hydroxyphenylpyruvic acid and their respective internal standards were separated after 10 µL injection using 0.1% acetic acid in MQ as mobile phase A and methanol as mobile phase B with the following gradient: 0–0.5 min, 1–18% B; 0.5–3.5 min, 18–25% B; 3.5–3.6 min, 25–100% B; 3.6–4.5 min, 100% B; 4.5–4.6 min, 100–1% B; 4.6–5 min, 1% B (method RPLC-3). Representative chromatograms for analyte separation are shown in Appendix A.

### 2.5. Mass Spectrometry

The separated analytes and internal standards were detected using an API 5000 MS/MS (AB Sciex, Framingham, MA, USA) in multiple reaction monitoring (MRM) mode. Ionization was performed using electrospray ionization–MS/MS in negative mode for indole-3-aldehyde, phenylpyruvic acid, 4-hydroxyphenylpyruvic acid and their internal standards, and in positive mode for the remaining analytes and internal standards. The MRM transitions corresponding to the molecular ions [M+H]^+^ and [M-H]^−^ used for analyte and internal standard detection, along with their retention times, are summarized in Appendix A.

Calibrators were measured in duplicate and quality control samples in triplicate. Samples were measured in duplicate in a randomized order. All plasma and ascites samples were measured within a single experiment to minimize experimental variation. All pleural effusion samples were measured within a separate experiment. Individual amino acid and metabolite concentrations are listed in Appendix A.

### 2.6. ELISA

IL4I1 levels were determined using the Human IL-4I1 DuoSet ELISA (DY5684-05; R&D Systems, Minneapolis, MN, USA) with the DuoSet ELISA Ancillary Reagent Kit 3 (#DY009; R&D Systems) as described in the manufacturer’s protocol. All samples were measured in duplicate. Individual IL4I1 concentrations are listed in Appendix A.

### 2.7. Statistical Analyses

Geometric means are presented in the graphs as a measure of central tendency since the amino acid, metabolite and IL4I1 levels follow a more lognormal rather than normal distribution.

The stability of amino acid and metabolite levels in blood stored at RT was evaluated using a one-way repeated measures ANOVA followed by Dunnett’s multiple comparisons test with the first timepoint as reference group. Significance of ANOVA *p*-values of the 15 performed tests was determined using the Benjamini–Hochberg procedure (FDR = 0.05). Unadjusted *p*-values are reported.

Differences between two sample groups (i.e., plasma of healthy donors versus ascites of ovarian cancer patients; or FIGO stage III versus stage IV disease) were analyzed using an unpaired, two-tailed Mann–Whitney U test. Paired differences between two sample groups (i.e., plasma versus ascites of ovarian cancer patients) were analyzed using a paired, two-tailed Student’s *t*-test using log-transformed concentrations. All significant differences were confirmed upon application of the Benjamini–Hochberg procedure (FDR = 0.05). Unadjusted *p*-values are reported.

Differences between three sample groups (i.e., plasma of healthy donors and plasma and ascites of ovarian cancer patients) were analyzed using a two-tailed, unpaired Kruskal–Wallis test followed by Dunn’s post hoc test. Since the data contain partially overlapping samples (i.e., *n* = 22 paired observations and, respectively, *n* = 2 and 10 unpaired observations for the ovarian cancer plasma and ascites groups), unpaired testing was performed after randomly assigning 15 of the 22 paired observations to the ovarian cancer plasma group, and the remaining 7 paired observations to the ovarian cancer ascites group, resulting in two groups of equal size (*n* = 17). Reported Kruskal–Wallis and post hoc *p*-values represent the 95th percentile of *p*-values obtained from repeated (k = 10,000) testing of datasets with randomly assigned observations.

Correlations between continuous variables were evaluated using Pearson’s correlation analysis using log-transformed concentrations. Values below the lower limit of quantification (LLOQ) were included at the LLOQ value in correlation analyses. Analysis of progression-free survival data was performed using the log-rank test with continuous variables split at the median and using univariate Cox regression analysis. Progression-free survival was defined as the duration of time between treatment completion and clinical disease progression. Correlation with overall survival was not evaluated, since 81% of the patients were still alive at the last follow-up.

A *p*-value ≤ 0.05 was considered to be statistically significant. All statistical analyses were performed in SPPS (version 27.0) or R (version 4.1.2).

## 3. Results

### 3.1. Patient Characteristics

Thirty-four patients diagnosed with advanced-stage (i.e., FIGO stage IIb–IV) high-grade serous ovarian cancer were included in the study (Table 1). The median age of the patients at the time of diagnosis was 62.5 years (interquartile range, IQR: 58–68 years). Two patients (5.9%) presented with stage II disease, 24 (70.6%) with stage III disease and eight (23.5%) with stage IV disease. Blood and ascites were collected at diagnosis from 24 and 32 patients, respectively, with 22 overlapping patients. Twenty-six patients (76.5%) completed treatment consisting of surgery and chemotherapy, and were followed-up for periods of 10 to 35 months. Clinicopathological characteristics and details on the treatment of these patients are listed in Appendix A.

A control group was formed by healthy female blood donors with a closely matched age range (median: 60.5 years; IQR: 55–65) to minimize age-related differences in amino acid metabolism between the groups. Ascites from patients with benign disease could not be collected in sufficient quantities during the study to constitute a direct control group for the malignant ascites samples.

### 3.2. Stability of Amino Acids and Metabolites in Blood Samples

Blood samples collected from ovarian cancer patients were processed into plasma the day after collection. Before processing, samples were stored at RT to allow for concurrent peripheral blood mononuclear cell (PBMC) collection (material not included in the present study). In contrast, blood from healthy donors and ascites from ovarian cancer patients were kept at 2–8 °C prior to processing.

To evaluate whether the plasma samples collected from the ovarian cancer patients could reliably be compared to the healthy donor plasma and patient ascites samples, the stability of amino acid and metabolite levels in whole blood samples stored at RT prior to plasma separation were evaluated by LC-MS/MS analysis. While the levels of most amino acids and metabolites remained stable over time, those of Arg, l-ornithine (Orn), l-glutamic acid (Glu), phenylpyruvic acid (PP) and 4-hydroxyphenylpyruvic acid (4HPP) were impacted by extended incubation at RT (Figure 2), a finding that was incorporated in the analysis of the patient samples as described below.

### 3.3. No Indication for Enhanced ARG1, iNOS or GLS1 Activity in Ovarian Cancer Patients

Evaluation of Arg and its metabolites Orn and l-citrulline (Cit) in whole blood kept at RT indicated a significant decrease in Arg and increase in Orn levels over time, whereas Cit levels remained stable (Figure 2A). Moreover, while Gln levels were unaffected by extended RT incubation, levels of its first metabolite Glu were significantly increased (Figure 2B). Comparison of Arg, Orn and Glu levels between healthy donors and ovarian cancer patients could therefore only be performed with their respective plasma and ascites samples, but not with plasma samples of the ovarian cancer patients (Figure 3). Although the composition of malignant ascites bears a certain degree of resemblance to that of plasma, these fluids are not identical [26] and direct comparison of plasma from healthy donors with ascites from ovarian cancer patients should therefore be considered with caution. Nonetheless, we believe that this comparison may still provide an initial indication of whether aberrant amino acid metabolism occurs in ovarian cancer patients.

Significantly higher Arg and lower Orn levels were found in the ascites of ovarian cancer patients compared to healthy donor plasma (Figure 3A). Moreover, Cit levels in both plasma and ascites of the patients were significantly lower than those in plasma of healthy donors (Figure 3A). Notably, these differences are opposite to those expected in the case of elevated ARG1 and/or iNOS activity in ovarian cancer. Furthermore, no significant difference in Gln or Glu levels was found between the patient samples and healthy donor plasma (Figure 3B). Overall, these observations do not provide an indication for enhanced ARG1, iNOS or GLS1 enzyme activity in high-grade serous ovarian cancer patients.

### 3.4. Elevated Trp Metabolism Is Dominated by IDO1/TDO—Rather Than IL4I1—Activity

Contrary to the poor stability of Arg, Orn and Glu levels in blood kept at RT (Figure 2A,B), Trp and its IDO1/TDO-catalyzed metabolite l-kynurenine (Kyn) showed remarkably stable levels over time (Figure 2C). Therefore, all three sample groups could be compared to evaluate IDO1/TDO-mediated Trp metabolism (Figure 4).

In the plasma of ovarian cancer patients, Trp levels were 2.2-fold decreased (based on geometric means) compared to those of healthy donors, although Kyn levels were not elevated accordingly (Figure 4A). Conversely, Kyn levels in the ovarian cancer ascites samples were on average 2.3- to 2.4-fold higher compared to those in the plasma samples, whereas Trp levels were still significantly lower than those in healthy donor plasma, but 1.4-fold higher compared to those in plasma of the patients (Figure 4A). Overall, significantly increased Kyn/Trp ratios were found in plasma of ovarian cancer patients compared to plasma of healthy donors (Figure 4B), and Kyn/Trp ratios in ovarian cancer ascites were even higher based on analysis of the paired samples (Figure 4C). Moreover, despite the distinct patterns of Trp and Kyn levels in the plasma and ascites samples of ovarian cancer patients (Figure 4A), a clear correlation was present between the Kyn/Trp ratios (Figure 4D). No significant correlations between Trp levels, Kyn levels or Kyn/Trp ratios and age, BMI, disease stage or progression-free survival (Appendix A) were found, except for a significant correlation between plasma Trp and BMI (Appendix A).

An alternative pathway of Trp metabolism, catalyzed by IL4I1, results in the formation of indole-3-pyruvic acid (I3P) (Figure 1). Inconveniently, we were not able to detect this metabolite by LC-MS/MS analysis, although this finding is in accordance with the previously reported instability of I3P [18,27,28]. As an alternative approach to evaluating IL4I1-mediated Trp metabolism, further downstream metabolites of Trp were considered as surrogate markers for I3P formation. Sadik and co-workers reported increased levels of indole-3-lactic acid (I3LA), indole-3-acetic acid (I3AA), indole-3-aldehyde (I3A) and kynurenic acid (KynA) in IL4I1-overexpressing cells compared to control cells [18]. All four of these downstream metabolites remained stable in blood stored at RT prior to plasma separation (Figure 2D) and were therefore evaluated in the healthy donor and ovarian cancer patient samples (Figure 5).

No significant differences in I3LA and KynA levels were found among the three sample groups (Figure 5). Moreover, while significant differences in I3AA and I3A levels were apparent, the levels of these metabolites were lower, rather than higher, in the ovarian cancer patient samples compared to healthy donor plasma (Figure 5). These results argue against an elevated metabolism of Trp by IL4I1 in high-grade serous ovarian cancer patients and point towards IDO1/TDO as the dominant source of enhanced Trp metabolism.

### 3.5. Enhanced IL4I1-Mediated Phe and Tyr Metabolism Correlates with Disease Stage

Although Trp metabolism by IL4I1 does not appear enhanced in high-grade serous ovarian cancer patients, this does not exclude an elevated metabolism of other substrates by IL4I1. Two other major substrates of IL4I1 are Phe and Tyr [19,29,30], which are converted into PP and 4HPP, respectively (Figure 1). In contrast to I3P, both of these metabolites could be detected by LC-MS/MS analysis, although their levels could not reliably be determined in the ovarian cancer plasma samples due to their instability in blood upon extended incubation at RT (Figure 2E,F).

PP and 4HPP levels in ovarian cancer ascites samples were 3.1- and 2.3-fold higher, respectively, compared to those in healthy donor plasma (based on geometric means), whereas Phe levels were only modestly higher and Tyr levels did not significantly differ between these groups (Figure 6A,B). Moreover, individual patients showed even up to 50-(PP) or 30-fold (4HPP) higher levels than the geometric mean of the healthy donor plasma samples (Figure 6A,B). These findings provide an indication that Phe and Tyr metabolism by IL4I1 may be enhanced in the ovarian cancer patients.

Although the levels of PP and 4HPP in the ovarian cancer plasma samples were affected by the sample processing conditions, a rough estimate of the originally present levels can still be made. Based on the estimated reductions in PP and 4HPP levels of 36% and 62%, respectively, after 24 h of incubation at RT (Appendix A), it can be deduced that the true levels in the ovarian cancer plasma samples would likely have been considerably lower than those found in the ascites samples (Figure 6A,B). Moreover, the PP and 4HPP levels in these plasma samples do not appear to be elevated compared to those in healthy donor plasma (Figure 6A,B), thus resembling the pattern observed for levels of the Trp-derived metabolite Kyn (Figure 4A).

PP and 4HPP levels in the ascites samples were strongly correlated to each other (Figure 6C), suggesting that they are generated by the same enzyme (i.e., IL4I1), rather than a combination of other enzymes capable of producing these metabolites. Since IL4I1 is a secreted enzyme, in contrast to IDO1 and TDO, its abundance could also directly be evaluated in the plasma and ascites samples. IL4I1 levels were detectable in an enzyme-linked immunosorbent assay (ELISA) in all ascites samples, and its expression was significantly correlated to both PP and 4HPP levels (Figure 6D). In contrast, the enzyme could not be detected in the majority of the plasma samples (i.e., within the detection range of the ELISA) (Appendix A), which is in accordance with the lower PP and 4HPP levels measured in plasma compared to ascites samples (Figure 6A,B).

Next, we tested if the PP and 4HPP levels in ovarian cancer ascites samples correlated with clinical parameters of the patients. No significant correlations were found with age, BMI or progression-free survival (Appendix A). However, both PP and 4HPP levels were significantly increased in patients with stage IV compared to stage III disease (Figure 6E). Moreover, while the difference in IL4I1 levels among the two disease stages was not found to be statistically significant, most stage IV patients with elevated PP and 4HPP levels also showed elevated IL4I1 levels (Figure 6E). These results indicate a potential role for IL4I1 in the progression of high-grade serous ovarian cancer through metabolism of Phe and Tyr, despite the apparent absence of elevated Trp metabolism by IL4I1.

### 3.6. Enhanced Phe and Tyr Metabolism by IL4I1 in Pleural Effusions of Lung Cancer Patients

To evaluate whether elevated metabolism of Phe and Tyr by IL4I1 is limited to ovarian cancer ascites, we analyzed pleural effusions from non-small cell lung cancer patients (Figure 7), which accumulate by similar mechanisms as ascites [31].

Pleural effusions were collected from 24 patients with advanced-stage (i.e., AJCC stage IV) non-small cell lung cancer. Similar to the ovarian cancer blood samples, pleural effusion samples were stored at RT prior to processing to allow for concurrent collection of tumor cells (material not included in the present study). Overall, lower PP and 4HPP levels were detected in the pleural effusion (Figure 7A) compared to the ascites samples (Figure 6A,B), despite the relatively similar IL4I1 levels (Figure 7B). As discussed for blood samples in Section 3.2, this is likely a direct consequence of the sample storage conditions. However, a subgroup of pleural effusion samples showed considerably higher PP and 4HPP levels (Figure 7A), indicating elevated metabolism in these patients. Similar to the ascites samples, PP and 4HPP levels in the pleural effusion samples were strongly correlated with each other (Figure 7C), while 4HPP levels also significantly correlated with the levels of IL4I1 (Figure 7D). These results demonstrate that elevated metabolism of Phe and Tyr by IL4I1 is not limited to ovarian cancer ascites, although the frequency among patients may differ depending on the type of cancer.

## 4. Discussion

Despite recent advances in the treatment of ovarian cancer patients [4], the potential clinical benefit of immunotherapy remains untapped due to the incomplete understanding of molecular mechanisms underlying immune suppression in ovarian cancer, including amino acid metabolism. We sought to investigate the role of amino acid-metabolizing enzymes in high-grade serous ovarian cancer by interrogating the abundance of relevant amino acids and metabolites in patient-derived liquid biopsies, including ascites as a promising source for biomarker identification. Our findings demonstrate enhanced amino acid metabolism by IDO1/TDO and IL4I1 in high-grade serous ovarian cancer patients, with markedly elevated metabolite levels in patient ascites samples compared to patient and healthy donor plasma. Notably, the prevailing metabolic profile of ovarian cancer patients was characterized by an apparent lack of elevated IL4I1 activity on Trp, the common substrate of IDO1, TDO and IL4I1. Nonetheless, elevated metabolism of Phe and Tyr by IL4I1 strongly correlated with disease stage, suggesting a potential role for IL4I1 in ovarian cancer progression.

For many years, IDO1 was viewed as one of the most promising targets for battling cancer through reactivation of the anticancer immune response. For this reason, IDO1 expression and activity has been evaluated in countless human cancer types [20], including ovarian cancer [32,33,34,35,36,37]. In contrast, a potential role for TDO in tumor immune escape is significantly less studied, particularly in cancers of the ovary [38,39]. In close agreement with previous findings [35,36,37], we report a two-fold increase in the Kyn/Trp ratio in ovarian cancer plasma compared to plasma of healthy donors, attributable to significantly decreased Trp levels. Moreover, we demonstrate for the first time that the Kyn/Trp ratio is even higher in the ascites of ovarian cancer patients, owing to concurrently lower Trp and higher Kyn levels. Despite unaltered plasma Kyn levels in ovarian cancer patients, the clear correlation between Kyn/Trp ratios in the plasma and ascites samples suggests that IDO1 and/or TDO activity is responsible for the metabolic changes in both fluids. Although these two enzymatic activities cannot be discriminated based on metabolomic analysis, the stronger mRNA expression of IDO1 compared to that of TDO in serous ovarian cancer tissues [18], and its considerably higher affinity and turnover rate for Trp [40], strongly point towards IDO1 as the major contributor. In the remainder of this article, we will hence specifically refer to IDO1 when discussing the elevated metabolism of Trp found in the high-grade serous ovarian cancer patients.

Unfortunately, inhibition of IDO1 as an anticancer strategy failed in a phase III clinical trial [41], significantly dampening the interest in IDO1 as a therapeutic target. The cause of these unsatisfactory results has been extensively debated in the literature, with possible explanations including suboptimal drug dosage, a lack of patient selection or stratification on the basis of IDO1 expression, and a compensatory role for TDO in the absence of IDO1 activity [42,43]. Recently, Sadik and co-workers proposed that IL4I1 expression may also underlie the resistance of patients against IDO1 inhibition, with activation of the AhR presented as their common mechanism of immune response blockade and promotion of tumor cell malignancy [18]. IL4I1 shows enhanced expression in a wide variety of tumor types [17,18], and its Trp-metabolizing activity yields various downstream metabolites which are associated with AhR agonism, including I3LA, I3AA, I3A and KynA [18,44,45,46]. At baseline disease, we found that the activity of IDO1 was increased in ovarian cancer patients compared to healthy controls, whereas elevated activity of IL4I1 was also found in ovarian cancer ascites samples. However, only IDO1 appeared to metabolize their common substrate, since levels of Trp downstream metabolites attributable to IL4I1 activity were not increased. Although it cannot be excluded that this observation may be related to the further metabolism of the Trp downstream metabolites, it also may reasonably be explained by an inability of IL4I1 to compete with IDO1 for their common substrate, as IL4I1 has a considerably lower affinity for Trp [40,47]. In contrast, IL4I1 may not experience significant competition from Phe- and Tyr-metabolizing enzymes, allowing efficient IL4I1-mediated conversion of these substrates. Upon inhibition of IDO1 in cancer patients, the amount of Trp available as a substrate for IL4I1 would likely be replenished. Since active IL4I1 is sufficiently expressed in ovarian cancer patients to modulate ascitic PP and 4HPP levels, this would likely allow the enhanced production of Trp-derived AhR agonists by IL4I1. In turn, this can result in maintenance of the immunosuppressive environment initiated by IDO1. The elevated activity of both IDO1 and IL4I1 found in ovarian cancer patients therefore supports the hypothesis of IL4I1 as a potential resistance mechanism against IDO1 inhibition.

Based on the elevated levels of PP and 4HPP found in the ascites of patients with stage IV compared to stage III disease, our data additionally indicate a correlation between IL4I1 activity and ovarian cancer progression. Similarly, IL4I1 mRNA expression is higher in patients with metastatic melanoma compared to those with primary melanoma [18]. In vitro, IL4I1 has been shown to promote the proliferation, migration and invasion of various tumor cell types [18,48], including ovarian cancer cells [49]. IL4I1 therefore presents a potential therapeutic target for ovarian cancer, while its expression, activity or metabolite levels may also serve as a biomarker for disease progression. Notably, metabolite levels associated with IDO1 and IL4I1 activity were elevated in ovarian cancer ascites compared to plasma samples. This indicates that ascites can be a valuable alternative to plasma for biomarker detection, which is in line with other studies investigating different types of biomarkers in this fluid [50,51]. The elevated metabolite levels in ascites may arise from the proximity of this fluid to the tumor as well as the presence of tumor and immune cells, likely expressing IDO1 and/or IL4I1, in the ascites itself. Alternatively, the further metabolism of these metabolites, or their transport towards tissues capable of this, may be less efficient in ascites compared to plasma. Metabolomic profiling of epithelial ovarian tumor biopsies has additionally demonstrated elevated levels of the same metabolites (i.e., Kyn, PP and 4HPP) when compared to normal ovary biopsies [52]. This indicates that ascites can also present a feasible, minimally invasive alternative to tumor biopsies for metabolomic analysis.

In addition to IDO1 and IL4I1, enhanced ARG1 expression has previously been reported in the context of human ovarian cancer [21,53,54]. In contrast, while iNOS and GLS1 overexpression has been demonstrated in various human cancer types [10,55], reports in ovarian cancer are sparse [56,57]. This is in line with the present findings that do not indicate elevated metabolism by iNOS or GLS1 in ovarian cancer patients. Moreover, in contrast to the reported ARG1 expression in ovarian cancer, indications for increased ARG1 activity were not found either. Instead, our findings of decreased Orn and Cit levels in the ovarian cancer patients correlate with other studies showing decreased levels of these metabolites in patient plasma or serum samples [58,59]. Together with the elevated Arg levels, these changes may signify enhanced Arg synthesis rather than metabolism, since Arg is (in)directly synthesized from Orn and Cit. Argininosuccinate synthase (ASS) and lyase (ASL) are responsible for Arg synthesis, and accordingly, increased ASS mRNA and protein expression has been found in epithelial ovarian tumors [60,61], while ASL overexpression has been reported in other cancer types [62,63].

The current study provides indications for a role of IL4I1 in high-grade serous ovarian cancer, although these findings should be validated in a larger patient cohort. A larger study could also identify correlations with patient outcome, which may have been missed in this study due to the limited number of patients that could be evaluated for (progression-free) survival and the relatively short follow-up time. Inclusion of patients with other types of ovarian cancer would additionally indicate the applicability of our findings to other ovarian cancer subtypes. Moreover, inclusion of patients with benign disease would further support the malignant nature of the elevated IL4I1 metabolism, while a direct comparison between metabolite levels in ascites and tumor biopsies from the same patients could substantiate the use of ascites over biopsies for biomarker evaluation. Finally, a lesson to be learned from this study is the importance of considering the stability of amino acid and metabolite levels in blood and other liquid biopsies during sample work-up and interpretation of results. When it is desired to use both plasma and PBMCs for experiments, blood is to be kept at RT prior to processing due to detrimental effects of refrigeration on PBMC recovery and viability [64]. Similarly, ascites and pleural effusion may require storage at RT for efficient isolation of viable tumor cells. However, extended storage at RT can be problematic for accurate metabolomics, as demonstrated here, in accordance with and in addition to previous reports [65,66]. These results underline the importance of efficient work-up of liquid biopsy samples and consideration of the potential effects of any delays or sub-optimal storage temperatures.

## 5. Conclusions

Our data suggest a role for both IDO1 and IL4I1 in high-grade serous ovarian cancer and indicate that IL4I1 may be involved in progression of the disease through metabolism of Phe and Tyr. The levels of metabolites produced by these enzymes were highest in ascites samples, revealing this fluid as a useful liquid biopsy source for biomarker detection. Further exploration of the implications of enhanced IL4I1 activity is warranted to pave the way for successful immunotherapeutic treatment of ovarian cancer patients.

## Figures and Tables

**Figure 1 cancers-15-00893-f001:**
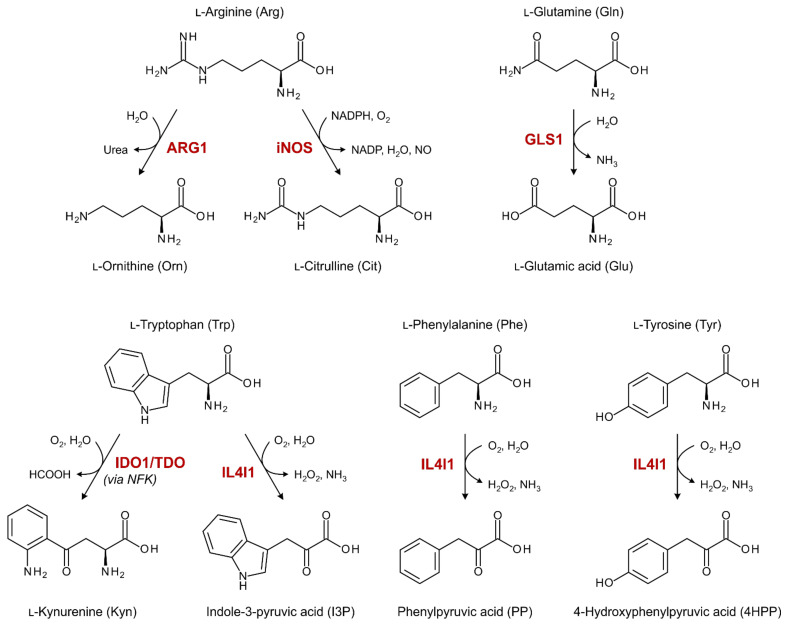
Overview of enzymatic reactions catalyzed by amino-acid metabolizing enzymes with known immunosuppressive properties in cancer. ARG1: arginase 1; GLS1: glutaminase 1; IDO1: indoleamine 2,3-dioxygenase 1; IL4I1: interleukin 4 induced 1; iNOS: inducible nitric oxide synthase; NFK: *N*-formylkynurenine; TDO: tryptophan 2,3-dioxygenase.

**Figure 2 cancers-15-00893-f002:**
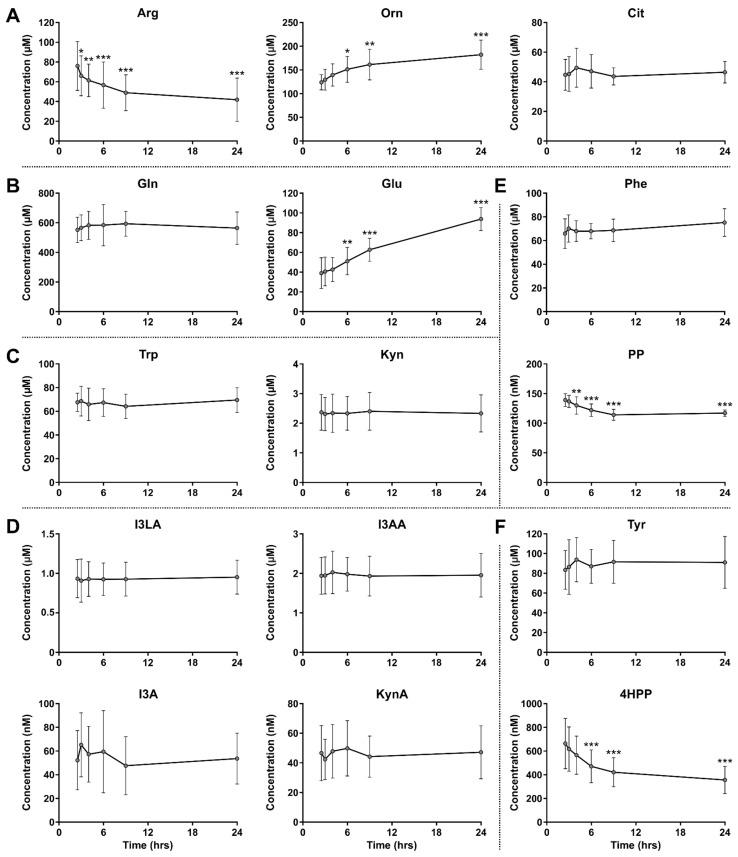
Stability of amino acid and metabolite levels in healthy donor blood stored at room temperature prior to plasma separation. Amino acids and metabolites related to metabolism of (**A**) Arg, (**B**) Gln, (**C**,**D**) Trp, (**E**) Phe and (**F**) Tyr. Amino acid and metabolite levels were determined by LC-MS/MS analysis and are expressed as mean ± SD of six individual donors. I3LA, I3AA, I3A and KynA are downstream metabolites of the unstable indole-3-pyruvic acid (I3P). One-way repeated measures ANOVA showed an overall effect of time on Arg, Orn, Glu, PP and 4HPP levels (all *p* < 0.001). Significant results of Dunnett’s multiple comparisons test with the first timepoint as reference group are indicated in the graphs. * *p* ≤ 0.05; ** *p* ≤ 0.01; *** *p* ≤ 0.001. 4HPP: 4-hydroxyphenylpyruvic acid; Arg: l-arginine; Cit: l-citrulline; Gln: l-glutamine; Glu: l-glutamic acid; I3A: indole-3-aldehyde; I3AA: indole-3-acetic acid; I3LA: indole-3-lactic acid; Kyn: l-kynurenine; KynA: kynurenic acid; Orn: l-ornithine; Phe: phenylalanine; PP: phenylpyruvic acid; Trp: l-tryptophan; Tyr: l-tyrosine.

**Figure 3 cancers-15-00893-f003:**
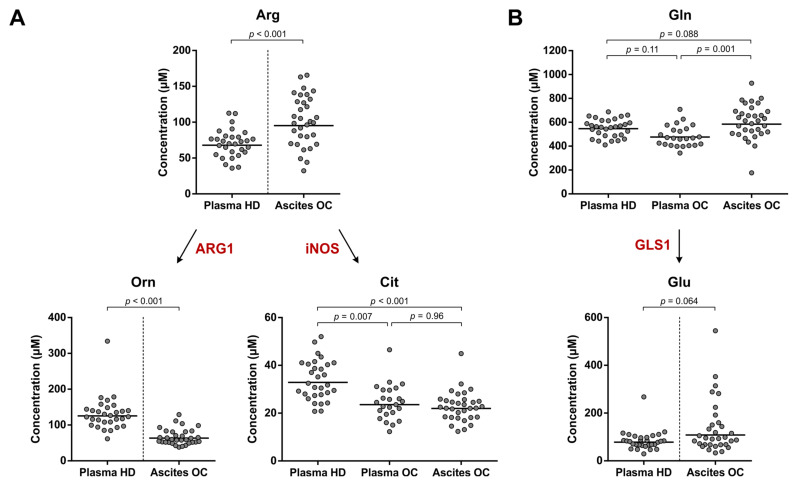
Arg and Gln metabolism in high-grade serous ovarian cancer patients and healthy donors. Amino acid and metabolite levels corresponding to (**A**) Arg and (**B**) Gln metabolism in plasma of healthy donors (HD) (*n* = 30) and plasma and ascites of ovarian cancer patients (OC) (*n* = 24 and 32, respectively, with *n* = 22 overlapping patients). The enzyme names indicate the enzymatic activities that are indirectly studied. Horizontal lines indicate geometric means. Repeated unpaired, two-tailed Kruskal–Wallis tests yielded significant results for comparison of Cit (*p* < 0.001) and Gln levels (*p* = 0.005) between the groups. Results of Dunn’s post hoc tests and unpaired, two-tailed Mann–Whitney U tests are indicated in the graphs. Arg: l-arginine; ARG1: arginase 1; Cit: l-citrulline; Gln: l-glutamine; GLS1: glutaminase 1; Glu: l-glutamic acid; iNOS: inducible nitric oxide synthase; Orn: l-ornithine.

**Figure 4 cancers-15-00893-f004:**
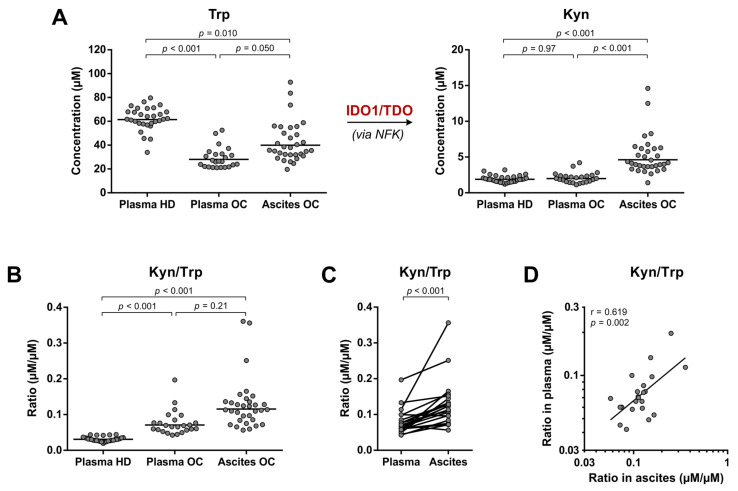
Trp metabolism towards Kyn formation in high-grade serous ovarian cancer patients and healthy donors. (**A**) Trp and Kyn levels and (**B**) Kyn/Trp ratios as indirect measures of IDO1/TDO enzymatic activity in plasma of healthy donors (HD) (*n* = 30) and plasma and ascites of ovarian cancer patients (OC) (*n* = 24 and 32, respectively, with *n* = 22 overlapping patients). Horizontal lines indicate geometric means. Repeated unpaired, two-tailed Kruskal–Wallis tests yielded significant results for comparison of Trp levels, Kyn levels and Kyn/Trp ratios between the groups (*p* < 0.001 for all). Results of Dunn’s post hoc test are indicated in the graphs. (**C**) Comparison and (**D**) correlation of Kyn/Trp ratios in paired plasma and ascites samples of ovarian cancer patients (*n* = 22). Results of a paired, two-tailed Student’s *t*-test and Pearson’s correlation analysis, both performed with log-transformed concentrations, are indicated in the graphs. IDO1: indoleamine 2,3-dioxygenase; Kyn: l-kynurenine; NFK: *N*-formylkynurenine; r: Pearson’s correlation coefficient; TDO: tryptophan 2,3-dioxygenase; Trp: l-tryptophan.

**Figure 5 cancers-15-00893-f005:**
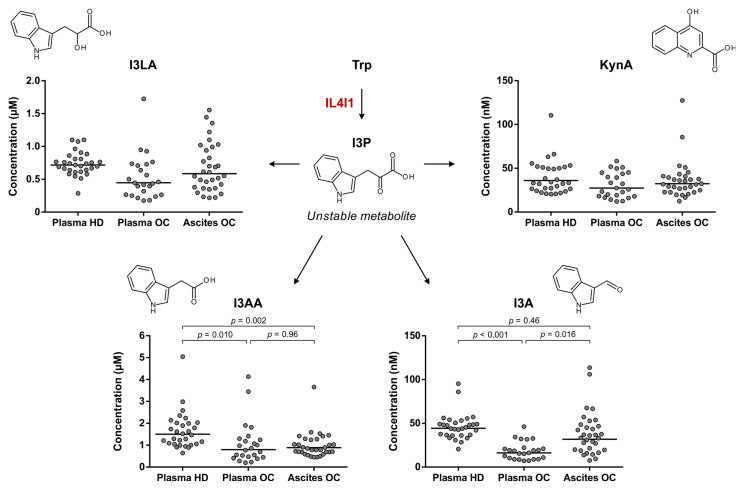
Trp metabolism towards I3P formation in high-grade serous ovarian cancer patients and healthy donors. Levels of Trp downstream metabolites are shown as surrogate measures of Trp-metabolizing IL4I1 activity in plasma of healthy donors (HD) (*n* = 30) and plasma and ascites of ovarian cancer patients (OC) (*n* = 24 and 32, respectively, with *n* = 22 overlapping patients). Horizontal lines indicate geometric means. Repeated unpaired, two-tailed Kruskal–Wallis tests yielded significant results for comparison of I3AA (*p* = 0.002) and I3A (*p* < 0.001)—but not I3LA and KynA—levels between the groups. Results of Dunn’s post hoc tests are indicated in the graphs where applicable. I3A: indole-3-aldehyde; I3AA: indole-3-acetic acid; I3LA: indole-3-lactic acid; I3P: indole-3-pyruvic acid; IL4I1: interleukin 4 induced 1; KynA: kynurenic acid; Trp: l-tryptophan.

**Figure 6 cancers-15-00893-f006:**
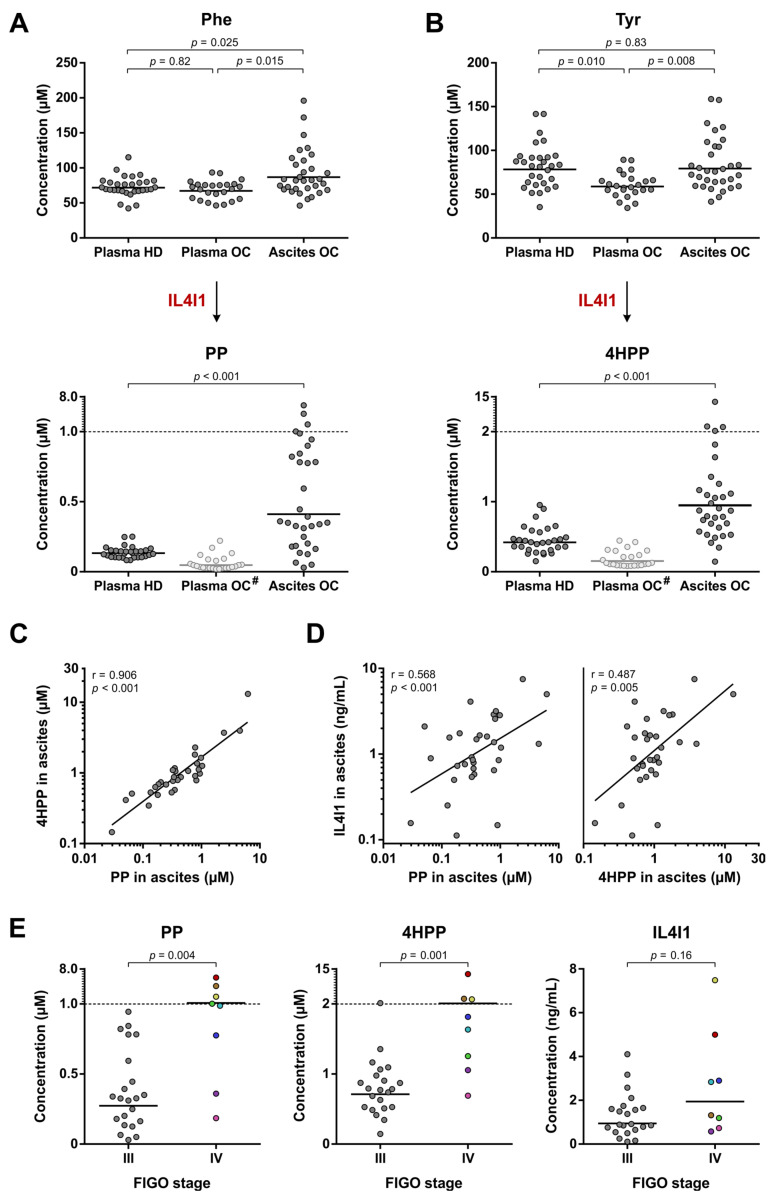
Phe and Tyr metabolism in high-grade serous ovarian cancer patients and healthy donors. Amino acid and metabolite levels corresponding to (**A**) Phe and (**B**) Tyr metabolism as indirect measures of IL4I1 enzymatic activity in plasma of healthy donors (HD) (*n* = 30) and plasma and ascites of ovarian cancer patients (OC) (*n* = 24 and 32, respectively, with *n* = 22 overlapping patients). Levels of Phe, Tyr, PP and 4HPP were determined by LC-MS/MS analysis. Horizontal lines indicate geometric means. Repeated unpaired, two-tailed Kruskal–Wallis tests yielded significant results for comparison of Phe (*p* = 0.025) and Tyr levels (*p* = 0.011) between the groups. Results of Dunn’s post hoc tests are indicated in the graphs. For comparison of PP and 4HPP levels between the two groups, results of the Mann–Whitney U test are indicated in the graphs. (**C**,**D**) Correlation between PP, 4HPP and IL4I1 levels in ascites of ovarian cancer patients (*n* = 32). Results of Pearson’s correlation analyses are indicated in the graphs. Levels of IL4I1 were determined by ELISA. (**E**) PP, 4HPP and IL4I1 levels in ascites of patients with stage III (*n* = 22) and stage IV disease (*n* = 8). Samples of patients with stage IV disease are colored individually to allow for comparison between the graphs. Results of unpaired, two-tailed Mann–Whitney U tests are indicated in the graphs. #: levels of PP and 4HPP in plasma of healthy donors are influenced by sample processing conditions and should therefore not directly be compared to those of the other sample groups; 4HPP: 4-hydroxyphenylpyruvic acid; FIGO: International Federation of Gynecology and Obstetrics; IL4I1: interleukin 4 induced 1; Phe: l-phenylalanine; PP: phenylpyruvic acid; r: Pearson’s correlation rank coefficient; Tyr: l-tyrosine.

**Figure 7 cancers-15-00893-f007:**
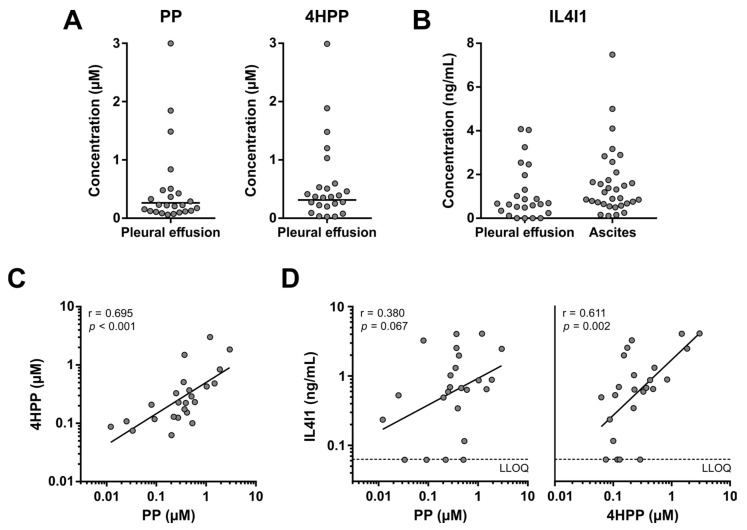
PP, 4HPP and IL4I1 levels in pleural effusion samples of patients with stage IV non-small cell lung cancer. (**A**) PP and 4HPP levels in pleural effusion samples (*n* = 24). Horizontal lines indicate geometric means. (**B**) IL4I1 levels in pleural effusion of non-small cell lung cancer patients (*n* = 24) and ascites of high-grade serous ovarian cancer patients (*n* = 32). IL4I1 levels in four pleural effusion samples were below the lower limit of quantification (LLOQ). Geometric means are therefore not displayed. (**C**,**D**) Correlation between PP, 4HPP and IL4I1 levels in pleural effusion samples. Dashed lines indicate the LLOQ. Samples with an IL4I1 level below the LLOQ are shown at the LLOQ value. Results of Pearson’s correlation analyses are indicated in the graphs. 4HPP: 4-hydroxyphenylpyruvic acid; IL4I1: interleukin 4 induced 1; PP: phenylpyruvic acid; r: Pearson’s correlation rank coefficient.

**Table 1 cancers-15-00893-t001:** Clinicopathological characteristics of included high-grade serous ovarian cancer patients and subgroups thereof.

		Subgroups Based on Collected Samples *		
	All Patients (*n* = 34)	Patients with Plasma Collected (*n* = 24)	Patients with Ascites Collected (*n* = 32)	Patients with Both Plasma and Ascites Collected (*n* = 22) ^†^
Age (years)				
Median (IQR)	62.5 (58–68)	62.5 (56–69)	63 (59–69)	63 (58–70)
BMI (kg/m^2^)				
Median (IQR)	24 (22–27)	24 (22–27)	25 (22–28)	24 (22–27)
FIGO stage				
II	2 (5.9%)	1 (4.2%)	2 (6.3%)	1 (4.5%)
III	24 (70.6%)	16 (66.7%)	22 (68.8%)	14 (63.6%)
IV	8 (23.5%)	7 (29.2%)	8 (25.0%)	7 (31.8%)
Primary treatment completed				
Yes	26 (76.5%)	20 (83.3%)	24 (75.0%)	18 (81.8%)
No	8 (23.5%)	4 (16.7%)	8 (25.0%)	4 (18.2%)

BMI: body mass index; FIGO: International Federation of Gynecology and Obstetrics; IQR: interquartile range. Percentages may not total to 100% due to rounding. * Samples from remaining patients were missing for logistic reasons. ^†^ Also referred to as “overlapping patients”.

## Data Availability

The data presented in this study are available in the Appendix A or upon request from the corresponding author.

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
