# Peer review of "Amino Acid-Metabolizing Enzymes in Advanced High-Grade Serous Ovarian Cancer Patients: Value of Ascites as Biomarker Source and Role for IL4I1 and IDO1"

_cancers, 2023, doi:10.3390/cancers15030893_

Round 1

Reviewer 1 Report

Summary

Grobben et al. report that the kynurenine pathway and the amino acid-metabolizing enzyme IL4I1 play an important role in ovarian cancer. Plasma samples of healthy donors and ovarian cancer patients were collected and the levels of amino acids and their metabolites were analyzed and compared. In addition, ascites samples from patients were analyzed. As plasma samples from patients were stored at room temperature for 24 h, Grobben et al. first evaluated the stability of amino acids and metabolites and detected significant changes of Arg, Orn, Glu, PP and 4HPP levels over time, making comparison of these metabolites with healthy plasma difficult. While Arg- and Gln-derived metabolites were not changed, Trp levels were significantly decreased in ovarian cancer patient samples. The Kyn/Trp ratio was significantly increased in patients’ plasma and was even higher in ascites samples. IL4I1-derived Trp metabolites were unaltered in the blood or ascites of ovarian cancer patients compared to healthy donors. IL4I1-derived metabolites from Tyr and Phe, 4HPP and PP were detected in ascites samples from ovarian cancer patients and correlated with each other as well as with IL4I1 protein levels. Moreover, IL4I1 protein and its products PP and 4HPP increased with FIGO stage in ovarian cancer. IL4I1 protein and PP/4HPP were also detected in pleural effusion samples from patients with NSCLC and also correlated with each other.

In general, the shown data is of interest. However, several points need to be clarified:

General points

Rather than focus on the comparison between plasma and ascites I would suggest to emphasize the presence of specific metabolites in ascites.

In this work, the main findings are related to the metabolism of aromatic amino acids, the authors should hence consider to focus on these findings.

The conclusion that Arg and Gln metabolism is not enhanced cannot be drawn as analysis of adjunct pathways was not performed (see below). I would hence suggest to move these data to the supplement and focus on the metabolism of aromatic amino acids.

Title

No correlation with disease stage is shown for IDO1, this is misleading in the title, please rephrase.

Introduction

The production of H2O2 by IL4I1 is mentioned in the introduction, please add that also NH3 is generated by IL4I1. You also show this in Fig. 1

Results

Fig. 2: The authors mention storage of patients’ blood samples at room temperature for several hours. How long were the other samples (ascites and blood from healthy donors) stored after collection? Were they stored at 2-8°C directly after collection? Metabolites can also degrade at 2-8°C. How long were they stored at this temperature? The stability of metabolites in ascites and blood samples from healthy donors stored in the same way as the samples used for the project should be tested corresponding to the experiment shown in Fig. 2.

For future experiments, it is highly recommended to directly freeze metabolic samples in liquid nitrogen.

Fig. 3: The authors explain the difficulty to compare plasma and ascites samples. However, these samples were compared by statistical analysis, which might lead to false interpretation of the data. This should be considered for all figures. Furthermore, the amino acids Arg and Glu play a role in various pathways, i.e. Arg is used for the biosynthesis of polyamines, while Glu is involved in the citrate cycle and the production of GABA, among others. It would be of great interest to check levels of other Arg- and Glu-derived metabolites.

As mentioned above, the authors may consider to focus on the metabolism of aromatic amino acids in ovarian cancer and mention the analysis of other metabolites in the supplements or omit them.

Fig. 5: As the authors explain, it is most likely that I3P and its derived metabolites are not produced in ovarian cancer. However, it might be possible that the metabolites are further metabolized and not released to plasma or ascites. This should be mentioned in the main text.

Fig. 6: As shown in Fig. 2, the stability of PP and 4HPP was affected by RT incubation and comparison not recommended. As in Fig. 3, please remove the measured concentrations of these metabolites in patient plasma. In addition, it should be considered to remove the statistical comparison of plasma HD and ascites in A and B but to focus on the detection of PP and 4HPP in ascites samples.

Discussion

 “Although these two enzymatic activities cannot be discriminated based on metabolomic analysis, the stronger mRNA expression of IDO1 compared to that of TDO in ovarian cancer tissues [18], and its considerably higher affinity for Trp [40], strongly point towards IDO1 as the major contributor.” This is highly speculative and could be omitted or rephrased a bit to make it sound less certain.

In line 546, the authors claim that the activity of IDO1 was increased in ovarian cancer patients. This is in contrast to the statement that the enzymatic activities of IDO1 and TDO “cannot be discriminated based on metabolomic analysis […]”. Please rephrase. This should be considered for the complete discussion (i.e. line 569).

Conclusion

The manuscript puts a lot of focus on IDO1. However, no data are shown that exclude a potential role of TDO2. Hence, I would suggest to add TDO2 as a potential Trp metabolic enzyme also in the title or refer to the kynurenine pathway that encompasses both.

Reviewer 2 Report

COMMENTS:

INTRODUCTION: Very well written. It contains a lot of relevant details that will be covered in the study.

RESULTS: Overall, the subtitles for each section should represent a key finding of the section to keep the readers more eager for the study.

Figure 3A: After how long at RT, the concentration of the metabolites was assessed? It is not indicated.

Line 384: data is always encouraged to be shown. Authors could add their correlation analysis for BMI in the supplementary results.

Figure 5: there is an error in the x-axe label for I3LA and KynA.

Figure 6: Please indicate in each letter, the type of assay performed. For example, for D, ELISA.

Figure 6E: The authors claim that the production of PP and 4HPP is mainly driven by IL4I1 activity. How authors could explain the not significant increase of IL4I1 enzyme in the IV OC stage when the products are significantly higher compared to stage III? What other enzymes could be inducing this significant increase if not IL4I1 at the IV stage?

Lines 470-471: Can the authors change the color of the dots for samples above the median in 6E so we can track for every metabolite the concentration and relevance in the specific patient samples with high concentration?  

Lines 471-473: since no significant increase is found for IL4I1, could authors reword these lines?

Figure 7A: can the authors plot the pleural effusion levels for PP and 4HPP along OC ascites samples as they did in 7B?

There are big limitations in this study:

1. Section 3.2. The extended incubation of the blood at RT affected the stability of 5 metabolites. How authors could exclude that their findings are relevant to normal physiology and not a consequence of extended incubation changing the total concentration of some metabolites and indirectly acting on others that weren’t necessarily measured?  

2. Lack of HD ascites. Plasma and ascites comparison is not relevant because depending on the stage of the disease the different concentrations of metabolites systemically (blood) would differ. Therefore, I do not feel confident drawing conclusions from a comparison like in figure 3 showing plasma from healthy donors and OC ascites. I would suggest adding an intersectional vertical dotted line inside figure 3 where there is only plasma from HD and OC ascites, to virtually separate the samples and keep aware the readers that this comparison is not accurate.

Requested/suggested experiments/analysis:

1. I suggest to the authors draw overall survival or progression-free survival plots by using https://kmplot.com/ for ovarian cancer and lung cancer and IL4I1 and the studied relevant metabolites to be able to make better conclusions for this study and a more general population of OC.

2. I suggest the authors perform an ELISA for IDO1/TDO to confirm elevated levels of this enzyme cause the elevated production of Kyn in OC ascites (and also to validate lines 400-402).

3. New conclusions/discussion to be addressed accordingly findings in suggestions 1 and 2. 

Round 2

Reviewer 1 Report

Fine for publication

Author Response

We thank the reviewer for their positive judgement of our manuscript and revisions.

Reviewer 2 Report

REVISED VERSION COMMENTS:

Thank you to the authors for kindly addressing and explaining all the previous comments/suggestions.

My only comment is about the title. The title should be more specific since the authors claim that they specifically studied advanced high-grade serous ovarian cancer samples. 

Author Response

We thank the reviewer for their positive judgement of our manuscript and revisions.

As suggested by the reviewer, we have amended the title to "Amino Acid-Metabolizing Enzymes in Advanced High-Grade Serous Ovarian Cancer Patients: Value of Ascites as Biomarker Source and Role for IL4I1 and IDO1"